# Design of Multiple Parallel-Arranged Perforated Panel Absorbers for Low Frequency Sound Absorption

**DOI:** 10.3390/ma12132099

**Published:** 2019-06-29

**Authors:** Xin Li, Qianqian Wu, Ludi Kang, Bilong Liu

**Affiliations:** School of Mechanical & Automobile Engineering, Qingdao University of Technology, No. 777 Jialingjiang Road, Qingdao 266520, China

**Keywords:** perforated panel, absorber array, low frequency absorption

## Abstract

A particular structure that consists of four parallel-arranged perforated panel absorbers (PPAs) is proposed for the low frequency sound absorption within a constraint space. The apertures of the perforated panels are set to ≥1.5 mm, and the number of orifices is much less and therefore easier to be produced in comparison with that of the micro perforated panel (MPP). A simple approximation model by using acoustic-electrical analogy is described to calculate the sound absorption coefficient of such device subject to normal wave incidence. Theoretical and experimental results demonstrate that the device can provide more than one octave sound absorption bandwidth at low frequencies.

## 1. Introduction

For an acoustical wave in the low frequency range, due to its large wavelength and super penetrating power, it is difficult to be handled by a thin structure. Micro perforated panel has been widely used as a wideband sound absorber replacing of traditional porous materials in circumstances requiring fireproof and environmental protection [1,2,3]. MPP absorber has sufficient acoustic resistance and small acoustic reactance provided by its sub-millimeter perforation, and can provide a bandwidth of one or two octaves near the resonance for sound absorption in comparison with other resonance based sound absorbers. One deficiency is that the MPP absorber requires larger cavity depth for low frequency sound absorption. Newly developed sound absorbers need to reduce the cavity depth while similar or better acoustical performance to the MPP absorber. Furthermore, although MPP absorber has tremendous potential for wide-band absorption up to 3 or 4 octaves when the aperture is set between 0.1–0.3 mm [4], the orifices are too many to be produced at low cost. For example, if a MPP absorber has aperture of 0.3 mm, the average sound absorption coefficient in the frequency range of 200–750 Hz could reach above 0.7. But an amount of 190,000 holes per square meter need to be punched on a panel with a thickness of 0.5 mm. Thus, wide application of MPP in low-frequency noise control is being hampered by high-cost manufacturing technology.

In recent years, for the purpose of expanding the absorption bandwidth, a series of extended structures based on MPP in parallel or series have been presented to introduce additional resonances. The combined structure of double or multi-layered MPPs and air cavities arranged in tandem along the direction of sound wave propagation were proposed successively [2,5,6,7,8]. The results show that the absorption bandwidth is extended to lower frequencies due to the additional resonance peak, while at the cost of increased air cavity. An alternately approach to extend the sound absorption bandwidth is to introduce multi-resonance through a compound MPP absorber array. Zha et al. [3] designed a parallel structure of the MPPs with two different cavities and measured its normal incidence absorption coefficients by impedance tube. They also explained that the appearance of two different resonance peaks is due to different cavity depths. Sakagami et al. [9] later proposed a combination of two different MPP absorbers for obtaining a wideband absorption device, such as the same MPPs with different air-cavity depths or different MPPs with the same air-cavity depth. Its absorption characteristics by considering the excess attenuation caused by the impedance discontinuity was described. The physical absorption mechanisms of an MPP absorber array consisting of three MPP absorbers with different cavity depths was investigated by Wang et al. [10]. They concluded that strong local resonance occurs due to different reactance matching conditions for the component MPP absorbers and the resonance frequencies shift due to inter-resonator interactions. Moreover, the oblique incidence sound absorption of parallel arrangement of four MPP absorbers in diffuse field was also investigated both numerically and experimentally [11]. Although the parallel MPP absorber array have the potential to improve sound absorption bandwidth due to multi-resonance of component MPPs, the reported perforation aperture is limited to 0.3–0.8 mm, and therefore the cost for the massive processing will be extremely high.

An elongated tube arrays of flexible tube bundles [12,13,14,15] or rigid extended tubes [16,17] attached to the perforated panel or MPP were designed to improve the low frequency sound absorption in a limited space. Among them, a perforated panel by parallel-arranged extended tubes (PPET) designed by Li et al. achieved more than one octave absorption bandwidth at low frequencies in a constrained space of 100mm. When the diameter of extend tubes were set at 2.9–5.1 mm and the length of extend tubes were 20–51 mm, the average sound absorption coefficient in the range of 120–250 Hz was more than 0.7. Although the perforation apertures of PPETs are much larger and the holes numbers are much less in comparison with that of the MPP absorbers, the lattice arrangement of a bundle of extend tubes makes the structure more complex and difficult to be manufactured.

In this paper, a composite structure of parallel-arranged perforated panel absorbers (PPAs) with unequal apertures of 1.5–4 mm is designed to improve low-frequency sound absorption in a limited space. The sound absorption performance of the designed absorber is almost equal to that of PPET in [16], and it is more practical because this device is simply a perforated panel of different apertures backed with same air cavity, without the extended tube bundle. As follows, a theoretical model of a particular structure consisting of four parallel-arranged perforated panel absorbers is described in Section 2, and the optimization parameters are given in in Section 3. After that, the test verification is shown in Section 4 and the conclusion is in Section 5.

## 2. Theoretical Model 

### 2.1. Acoustic Impedance of the Perforated Panel

A perforated panel is regarded as a parallel connection of numerous millimeter-level tubes with a certain depth. Therefore, the acoustic impedance of the perforated panel is equal to the acoustic impedance of a single tube divided by its perforation ratio. At first, Rayleigh studied the sound wave propagation in the tube, Crandall later simplified the sound wave propagation for short tubes [18]. When the diameter and length of the tube are far less than the wavelength of sound wave, the specific acoustic impedance of the single tube is defined as
(1)$$z=\frac{\Delta P}{\bar{u}}=j\omega \rho t{\left[1-\frac{2}{k\sqrt{-j}}\frac{J_{1}(k\sqrt{-j})}{J_{0}(k\sqrt{-j})}\right]}^{-1}$$
where ΔP is the sound pressure difference between the two ends of the tube, $\bar{u}$ is the average velocity across the cross-section of the tube, $k=d/2\sqrt{\omega \rho /\eta }$ is the ratio of the inner radius to the viscous boundary layer thickness inside the tube, ω=2πf is the angular frequency, ρ is the air density, d and t are the diameter and length of the tube.

For Equation (1), Maa [1] further gave the approximate formula of the acoustic impedance of the tube for all k values, which is expressed as
(2)$$z=\frac{32\eta t}{d^{2}}{\left(1+\frac{k^{2}}{32}\right)}^{1/2}+j\omega \rho t\left(1+{\left(9+\frac{x^{2}}{2}\right)}^{-1/2}\right)$$

For this design, the perforated panel of a certain thickness is assumed to be acoustically rigid, and the vibration effect of the plate under acoustic load is ignored. In addition, due to the acoustic radiation at both ends of the tube and the air flow friction on the surface plate, the correction of the acoustic resistance and the acoustic mass of the tube should be added to Equation (2). Therefore, the normalized characteristic impedance for the perforated panel with the perforation ratio *p* is written as
(3)$$Z_{PP}=\frac{32\eta t}{p\rho cd^{2}}\left(\sqrt{1+\frac{k^{2}}{32}}+\frac{\sqrt{2}}{32}k\frac{d}{t}\right)+\frac{\omega t}{pc}\left(1+1/\sqrt{3^{2}+\frac{k^{2}}{2}}+0.85\frac{d}{t}\right)$$

### 2.2. Acoustic Impedance of the Parallel-Arranged Perforated Panel Absorber

The perforated panel absorber array consists of four sub-PPA arranged in parallel, its structure diagram is shown in Figure 1a. Using the electrical equivalent circuit model in Figure 1b, this perforated panel absorber is regarded as a parallel combination of four RLC branch. Moreover, the back cavity is divided into four sub-cavities by the rigid clapboards to avoid the interaction between the sub-perforated panel resonators. Then, the normalized acoustic impedance for each sub-perforated panel absorber is expressed as
(4)$$Z_{i}=r_{i}+j\omega m_{i}-j\cot\left(\omega D_{i}/c\right)$$
(5)$$r_{i}=\frac{32\eta t_{i}}{p_{i}\rho cd_{i}{}^{2}}k_{r_{i}},k_{r_{i}}=\sqrt{1+\frac{k_{i}{}^{2}}{32}}+\frac{\sqrt{2}}{32}k_{i}\frac{d_{i}}{t_{i}}$$
(6)$$\omega m_{i}=\frac{\omega t_{i}}{pic}k_{m_{i}},k_{m}{}_{{}_{i}}=1+{\left(3^{2}+\frac{k_{i}{}^{2}}{2}\right)}^{-1/2}+0.85\frac{d_{i}}{t_{i}}$$

The total acoustic impedance of the four parallel-arranged PPAs is written as
(7)$$Z={\left(\sum_{i=1}^{4}\frac{\phi_{i}}{Z_{i}}\right)}^{-1}$$
where the subscript i=1,2,3,4 denotes the number of sub-perforated panel, $p_{i}$ is the perforation ratio of each sub-perforated panel, η is the viscous coefficient of the air, c is the speed of sound wave in air and ρc is the characteristic impedance in air, $r_{i}$ and $m_{i}$ are the normalized acoustic resistance and reactance of sub-perforated panel respectively, $t_{i}$ is the thickness of each sub-perforated panel, $D_{i}$ is the depth of the back cavity of each sub-perforated panel.

Then, for the normal incidence condition, the sound absorption coefficient of the parallel-arranged PPAs is expressed as
(8)α=4Real(Z)(1+Real(Z))2+(Imag(Z))2

## 3. Model Optimization and Comparison

Simulated annealing is an effective, fast and straightforward general probabilistic algorithm, which is widely used to obtain the best possible configuration for sound absorption systems due to its global optimization [19,20]. Similarly, in this paper, the parameters of sound absorption structure are optimized by simulated annealing method to seek the maximum average sound absorption within a specific frequency range.

### 3.1. Comparison of Sound Absorption between Perforated Panel and MPP in Low Frequencies

In the frequency range of 100–300 Hz, the sound absorption of single perforated panel (d = 0.7 mm) and MPP (d = 1.5 mm) with the same ratio (t/d = 1.3) are plotted in Figure 2. By contrast, it can be seen that sound absorption coefficient of the perforated panel is significantly higher than that of a single MPP (d = 0.7 mm), and the resonance peak moves to the lower frequency. The perforated panel has a higher sound absorption coefficient because the acoustic resistance of the perforated panel matches the characteristic impedance of air by reasonably adjusting the ratio of the larger aperture and plate thickness. From Figure 2b, it can be confirmed that the normalized acoustic resistance of the MPP is much higher than 1 because of the overdamping caused by the tiny aperture, while that of the perforated panel is close to 1 in the range of 100–200 Hz. In addition, Figure 2c shows that the normalized zero acoustic reactance of the perforated panel occurs at 185 Hz, and the normalized zero acoustic reactance of MPP occurs at 220 Hz. This intuitively explains that the resonance frequency of the perforated panel is lower than that of MPP. In other words, the shift of resonance to lower frequency is mainly attributed to the increase of acoustic mass with the increase of aperture depth.

However, an obvious defect is that the effective sound absorption bandwidth of single perforated panel is narrow. Therefore, a combined structure of multiple perforated panels in parallel needs to be designed and optimized to expand the sound absorption bandwidth for the low frequencies.

### 3.2. Comparison of Sound Absorption between PPAs and Other Resonant Structures in the Same Depth Cavity

In order to compare the sound absorption performance of the parallel-arranged PPAs and the existing resonant absorption structures in the same frequency range, the maximum average sound absorption coefficient in the range of 120–250 Hz was selected for comparison. Table 1 lists optimized parameters of the parallel-arranged PPAs, PPETs [16] and MPP absorber. The sound absorption coefficients of the three comparative structures are plotted in Figure 3.

Due to the nature of its own Helmholtz resonator, the absorption bandwidth of MPP occurs near the resonance, while the absorption deviating from the resonance drops rapidly. The parallel-arranged PPAs extends the absorption bandwidth through four resonances. For the parallel-arranged PPAs, four resonances and three anti-resonances occur due to each sub-PPA as a resonator and its coupling with each other. As shown in Figure 3, the four resonant frequencies occur at 130 Hz, 160 Hz, 190 Hz, and 230 Hz, and their sound absorption coefficients are 0.87, 0.96, 0.96, and 0.96 respectively. Moreover, the effective sound absorption bandwidth of the parallel-arranged PPAs is about 130 Hz (the sound absorption coefficients of 121–251 Hz are higher than 0.7). In the same restricted space of D = 100 mm, the sound absorption of the parallel-arranged PPAs is significantly better than that of a single MPP in the frequency ranges of 110–165 Hz and 215–250 Hz. That is, in comparison with that of a single MPP, the improvement of the bandwidth is about 55 Hz under 165 Hz and about 45 Hz above 215 Hz. Therefore, this proposed structure can obviously extend the sound absorption bandwidth compared with a single MPP in the same frequency range. 

In the range of 120–250 Hz, the sound absorption performance of the parallel-arranged PPAs is close to that of PPETs absorber [16]. Although the sound absorption coefficients of the first resonance and anti-resonances are slightly lower than that of PPETs, they are greater than 0.7, especially the transition between anti-resonance and resonance becomes flatter as the frequency tends to be higher. Besides, by comparing geometric parameters, the aperture of the parallel-arranged PPAs is between 2–3.1 mm, and that of PPETs is between 2.9–5.1 mm. Even though the aperture of the parallel-arranged PPAs is smaller than that of PPETs, its depth diameter ratio is much lower than that of the latter. Considering the processing technology, the hole with smaller depth diameter ratio is more convenient to be processed by common machining. Also, the perforation rate of the perforated panel absorber is lower than that of the PPETs. As for the parallel-arranged PPAs, about 57 holes per square decimeter are drilled on a panel; while for the PPETs, about 193 thin hollow tubes of different lengths per square decimeter are attached to a perforated panel in parallel. So, the designed structure is not complicated and simple to manufacture. 

### 3.3. Comparison of PPAs with Different Cavity Depth 

Figure 4 shows the sound absorption optimization of the four parallel-arranged PPAs with different cavity depth. The four solid lines (from left to right) represent the optimized sound absorption of four different cavities of 100 mm, 70 mm, 50 mm and 35 mm, corresponding to four different frequency bands, such as 200–450 Hz, 300–550 Hz, 400–650 Hz, and 400–750 Hz respectively. And the sound absorption coefficient of each peak and peak valley is above 0.85. The effective absorption bandwidth is about 3–4 1/3 octaves. Besides, the apertures of the PPAs ≥ 1.5 mm and the maximum thickness of the plate is 2.5 mm, as listed in Table 2. Therefore, when the aperture is designed to be ≥1.5 mm, this four parallel-arranged PPAs structure can be applied to different frequency bands for sound absorption at low frequencies by adjusting the depth of the back cavity and optimizing structural parameters.

## 4. Experimental Validation

The proposed structure can be made of metallic materials (stainless steel, aluminum) or non-metallic materials such as plastic wood and etc. The aperture of the PPAs is not less than 1.5mm and the maximum depth of the hole is 2.5 mm. For stainless steel or aluminum, mechanical or laser drilling can be used for bulk processing. The focus of this research is on the configuration and implementation of the structure, thus, a perforated plate made of resin produced by 3D technology is considered for verification.

The measurements are carried out in an impedance tube (SW422) according to ISO 10534-2 [21], and the instruments used in the experiment are shown in Figure 5. Based on the transfer function method, this device measures the sound pressure at two locations using two microphones and calculates the normal incident sound absorption coefficient. When two microphones connected to the data acquisition instrument are installed at position 0 and position 2 of the impedance tube, the sound absorption coefficient of 63–500 Hz is measured; when two microphones are placed in position 1 and position 2, the absorption coefficient of 400–1600 Hz is measured. The samples in the experiment are composed of perforated panel, clapboard and cylindrical shell, and an integrated structure of resin materials produced by 3D technology. The thickness of clapboard and cylindrical shell is 2mm and 2 mm, respectively. Besides, the sample must be sealed and mounted at the back of the impedance tube to ensure the presence of multiple resonance peaks.

In order to illustrate the sound absorption characteristics of the perforated panel absorber in different frequency range, two structures with different parameters are made for experimental verification. Sample A with aperture of 2.1mm, 2.5mm,3 mm is used for 120–300 Hz, and sample B with aperture of 1.5 mm,1.6 mm and 2.2 mm is used for 150–450 Hz.The parameters of samples are listed in Table 3 and Table 4, respectively. 

As plotted in Figure 6 and Figure 7, it can be seen that the theoretical prediction is in a relatively good agreement with the experimental results. The measurement results in Figure 6 show that sample A has an effective sound absorption bandwidth of 140 Hz within 135–275 Hz, and this compact structure can absorb large wavelength when the thickness is only 1/27 of the incident wavelength. Besides, the measured absorption coefficient of sample B exceeds 0.6 from 155–455 Hz, and the effective absorption bandwidth is extended to about four 1/3 octaves, as shown in Figure 7. Therefore, it is further confirmed that this designed perforated panel absorber has great potential in extending the sound absorption bandwidth at low frequency.

## 5. Conclusions

A parallel-arranged perforated panel absorber for low frequency sound absorption is investigated. A theoretical model is described to predict the acoustic properties. The sound absorption performance of the parallel-arranged PPAs with that of the existing resonant structure is discussed, which provide the basis to illustrate the feasibility of the designed structure. Samples for two frequency bands of 120–250 Hz and 150–450 Hz are designed and measured. The effective sound absorption covers the frequency range from 135 Hz to 275 Hz and 155–455 Hz respectively. Theoretical and experimental results prove that a wideband sound absorption of more than one octave can be achieved by a parallel combination of perforated panels with same cavity depth at low frequencies. Further work is to study the sound absorption performance of perforated panels with different cavity depth for low-frequency acoustic absorption and the perforated panel array in diffuse field.

## Figures and Tables

**Figure 1 materials-12-02099-f001:**
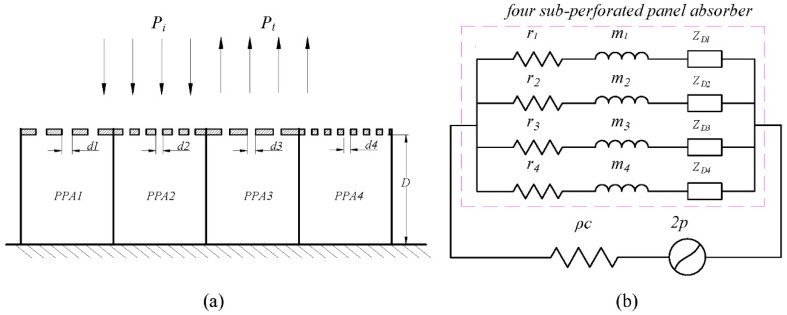
Schematic diagram and electrical equivalent circuit model: (**a**) Schematic of the four parallel-arranged perforated panel absorbers. (**b**) The electrical equivalent circuit model of the four parallel-arranged perforated panel absorbers.

**Figure 2 materials-12-02099-f002:**
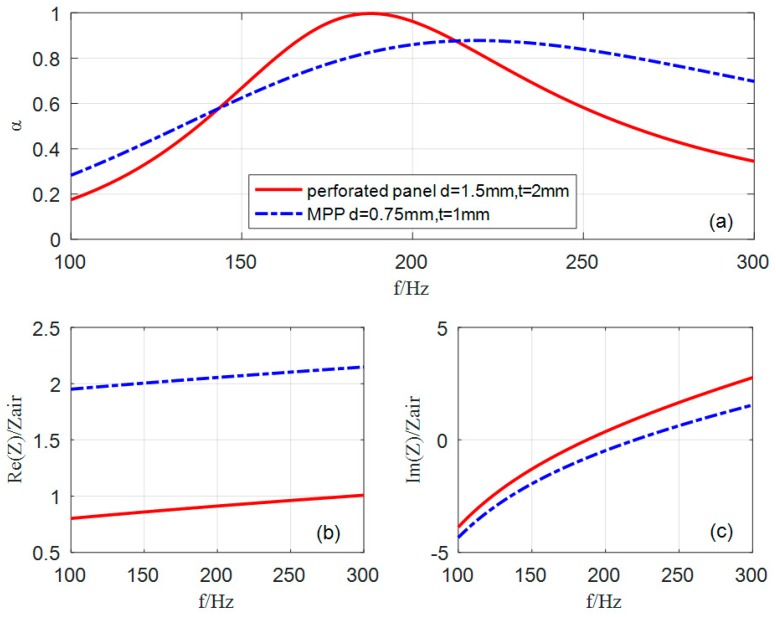
Comparison of the sound absorption of single perforated panel and MPP: (**a**) sound absorption coefficient; (**b**) normalized acoustic resistance; (**c**) normalized acoustic reactance. The parameters of the single optimized perforated panel are as follows: d = 1.5 mm, t = 2 mm, D = 100 mm and p = 0.46%; the parameters of the single optimized MPP are as follows: d = 0.75 mm, t = 1 mm, D = 100 mm and p = 0.33%.

**Figure 3 materials-12-02099-f003:**
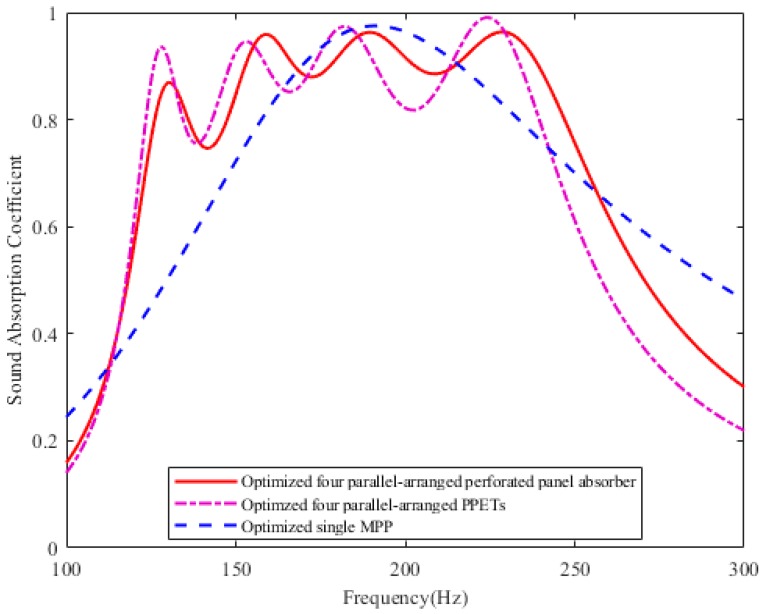
Comparison of the sound absorption of the four parallel-arranged PPAs and PPETs, MPP absorber in the frequency range of 120–250 Hz.

**Figure 4 materials-12-02099-f004:**
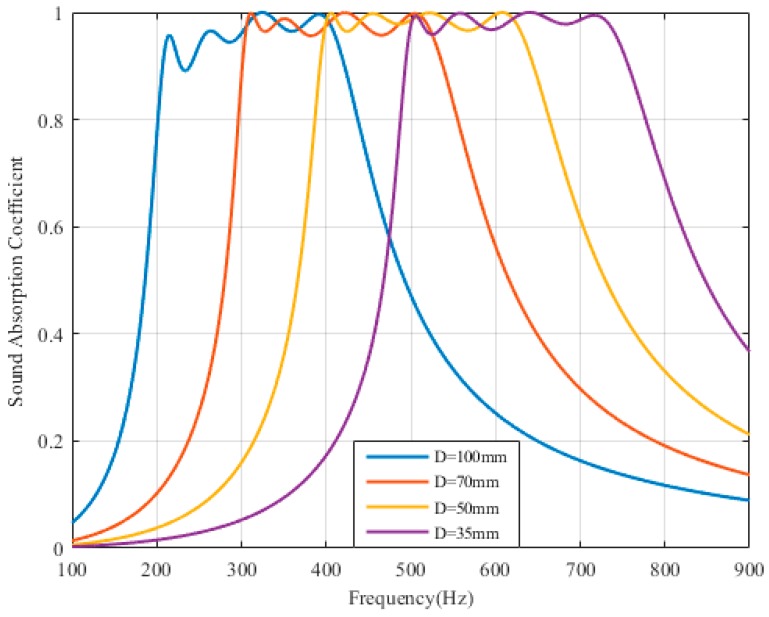
Optimization of the parallel-arranged PPAs with different cavity depth.

**Figure 5 materials-12-02099-f005:**
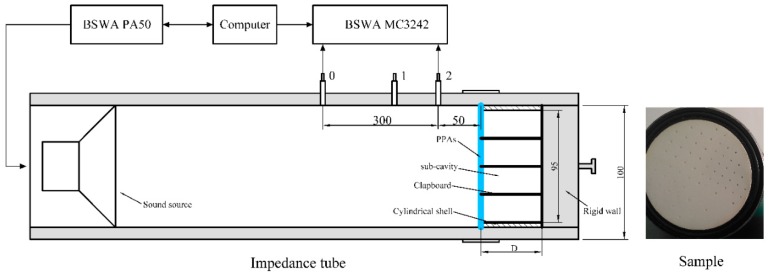
The experimental Setup.

**Figure 6 materials-12-02099-f006:**
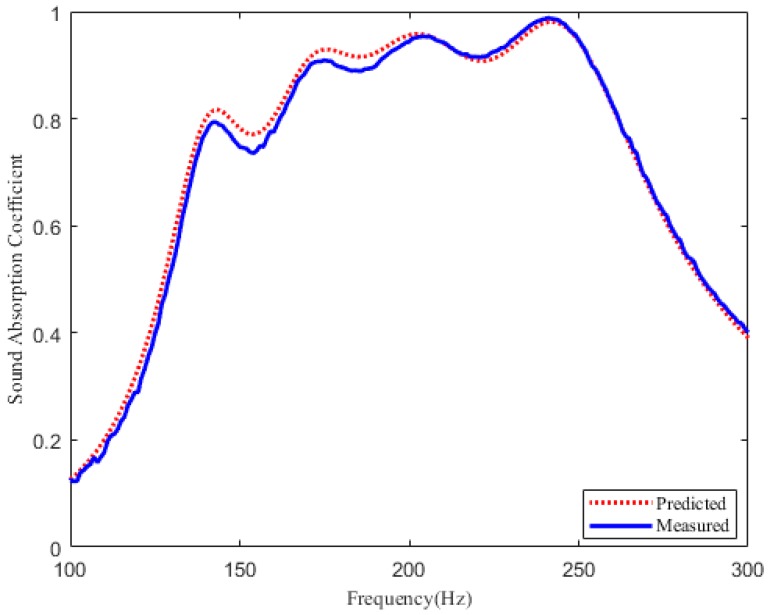
Sound absorption coefficient of the four parallel-arranged PPAs for 125–300 Hz. Red dotted line: predicted results. Blue solid line: measured results.

**Figure 7 materials-12-02099-f007:**
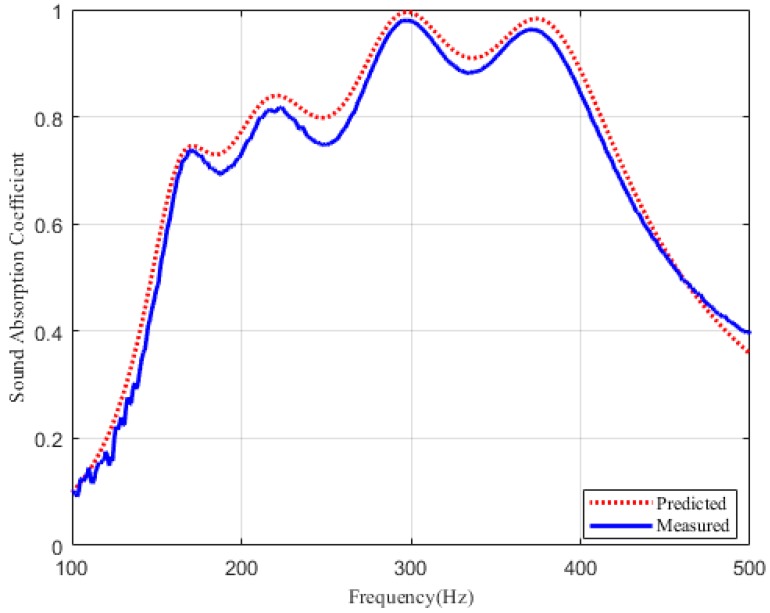
Sound absorption coefficient of the four parallel-arranged PPAs for 150–450 Hz. Red dotted line: predicted results. Blue solid line: measured results.

**Table 1 materials-12-02099-t001:** Optimization parameters of three sound absorption structures.

Parameter	PPAs				PPETs [16]				MPP
	PPA_1_	PPA_2_	PPA_3_	PPA_4_	PPET_1_	PPET_2_	PPET_3_	PPET_4_	
d (mm)	3.1	2.8	2.2	2	5.1	3.5	3.1	2.9	0.8
p (%)	0.32	0.46	0.62	0.93	3.26	3.74	5.31	4.43	0.41
t (mm)	2.5	2.5	2.5	2.5	51	40	40	20	2
D (mm)	100	100	100	100	100	100	100	100	100

**Table 2 materials-12-02099-t002:** Optimization parameters of PPAs with different cavity depth.

**Parameter**	**D = 100 mm**				**D = 70 mm**			
	**PPA_1_**	**PPA_2_**	**PPA_3_**	**PPA_4_**	**PPA_1_**	**PPA_2_**	**PPA_3_**	**PPA_4_**
d (mm)	2.4	1.5	1.5	1.6	3.5	1.5	1.6	1.5
p (%)	0.74	2.94	1.63	1.01	1.20	1.79	1.18	3
t (mm)	2.3	2.3	2.3	2.3	2.1	2.1	2.1	2.1
**Parameter**	**D = 50 mm**				**D = 35 mm**			
	**PPA_1_**	**PPA_2_**	**PPA_3_**	**PPA_4_**	**PPA_1_**	**PPA_2_**	**PPA_3_**	**PPA_4_**
d (mm)	1.7	3.1	1.5	1.5	1.8	1.5	1.5	4
p (%)	1.44	1.39	3	1.98	1.66	3	2.19	1.81
t (mm)	2.2	2.2	2.2	2.2	2.5	2.5	2.5	2.5

**Table 3 materials-12-02099-t003:** Parameters of the four parallel-arranged PPAs for 120–250 Hz.

Sample A	d (mm)	p (%)	t (mm)	D (mm)
PPA_1_	3	0.36	2.5	100
PPA_2_	2.5	0.50	2.5	100
PPA_3_	2.1	0.64	2.5	100
PPA_4_	2.1	0.96	2.5	100

**Table 4 materials-12-02099-t004:** Parameters of the four parallel-arranged perforated panel absorbers for 150–450 Hz.

Sample B	d (mm)	p (%)	t (mm)	D (mm)
PPA_1_	2.2	0.39	2	100
PPA_2_	1.6	0.63	2	100
PPA_3_	1.6	2.52	2	100
PPA_4_	1.5	1.26	2	100

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
