# Peer review of "Design of Multiple Parallel-Arranged Perforated Panel Absorbers for Low Frequency Sound Absorption"

_materials, 2019, doi:10.3390/ma12132099_

Round 1

Reviewer 1 Report

The subject, the theoretical model and the comparison with measurement are interesting and the results could be helpful to the designers of noise control.

The paper needs only editorial revisions.

See attached file for detailed remarks.

Author Response

Detailed response to comments by reviewer:

Query: Detailed remarks:

Point 1: Page 1 line 19: Write in full the meaning of MPP (MPP = Multiple Parallel Perforated panel?) .

Point 2: Page 2 line 63: correct “difficulty”.

Point 3: Page 2 line 67: delete “Ref”.

Point 4: Page 2, 80: in eq. (1) the term “u” is missing.

Point 5: Page 3, line 97 : label of Figure 1: insert colon “:” between model and (a).

Point 6: Page 3, line 101: correct “its structure diagram are (?) shown”

Point 7: Page 4, line 137: delete “is” between “coefficient” and “because”.

Point 8: Page 5, lines 163-166: rewrite more clearly.

Point 9: Page 6, line 189: insert “in” before Figure 4.

Point 10: Page 6, line 190: delete “it is” before “ an integrate structure…”.

Point 11: Page 8 line 224: rewrite with better English “Further work is the design of…”.

Reply:

Thank you for your comments. Corrections have been made according to the suggestions. 

Reviewer 2 Report

The presented paper “Design of multiple parallel-arranged perforated  panel absorbers for low frequency sound absorption” reports an investigation on low frequency sound absorption impedance tube measurements and simple approximation models. The following issues emerge:

The authors do not clarify how their work goes beyond the existing literature and does not clarify the aim of the current experiments.

Their thesis is mainly supported by the cost of the manufacturing technology, but they do not clarify how their proposal overcomes this limitation. At what extent? How is this quantifiable?

The terminology used eg. MPP, PPETs etc should be given systematically and should be clarified giving design parameters.

How can be low frequencies measured in an impedance tube? How are these data reported to the scale of a panel that is used in-situ or commercialized? There are a number of studies that report significant differences at low frequencies.

A clear discussion and conclusions of the results is missing. 

References should be written with more accuracy. 

Author Response

General reply to three reviewers´ comments:

In our opinion the merits of our manuscript are:

1) The designed absorber introduces multiple resonances to expand the absorption bandwidth of more than one octave. In the restricted cavity space, the bandwidth of the proposed structure is obviously wider than that of MPP. Besides, the structure proposed avoids the extend tubes in comparison with that of PPETs when the similar sound absorption performance is concerned.

2)  The aperture of the structure designed for low frequency sound absorption is much larger in comparison with the MPP and therefore has much lower perforation numbers.

3)  The depth of each sub-cavity of the designed structure has the same size, which is convenient for production in practice.

Changes made:

1) To illustrate the potential of the extended bandwidth of the design structure, Section 3.2 has been revised and expanded.

2) In order to show that the design structure can be applied to different frequencies, Section 3.3 has been added.

3) A section discussing “how the panels could be made in practice, which materials could be used” been added to Section 4, as lines: 205-210.

4) Low-frequency absorption measurement in an impedance tube has been added in Section 4, as lines: 211-217, and the setup diagram has been revised, as Figure5.

5) After repeated testing and correction, the measurement results are in good consistency with the theoretical results. Figure 6 and Figure7 has been revised.

6) The references have been revised.

Detailed response to comments by reviewer:

Query: The authors do not clarify how their work goes beyond the existing literature and does not clarify the aim of the current experiments.

Reply: Thank you for your comments. The main purpose of the paper is that the four parallel-arranged perforated panel absorbers (apertures ≥ 1.5mm) designed to expand absorption bandwidth at low frequencies. Section 3.2 is modified and illustrates the comparison between the present work and the existing literature. In the same restricted space of D=100mm, the sound absorption of the parallel-arranged PPAs is significantly better than that of a single MPP in the frequency ranges of 110-165Hz and 215-250Hz. Compared with PPETs, the designed perforated plate absorber also introduces multi-resonance to expand the absorption bandwidth, and its bandwidth is wider than that of the latter.

Query: Their thesis is mainly supported by the cost of the manufacturing technology, but they do not clarify how their proposal overcomes this limitation. At what extent? How is this quantifiable?

Reply: The MPP absorber was first proposed by Maa in 1970s, and since then the MPP absorber has been used widely as a wideband sound absorber.  One of the key factor affects the absorption coefficient of the MPP is the diameter of the perforation holes. To achieve optimized sound absorption, the aperture of the MPP hole should have the order of decimillimetre. In practical application, however, if the ratio of the aperture to the plate thickness is less than one for a typical metallic plate, the cheapest mechanical perforation technology is not applicable due to insufficient hardness of punch pins (Approximately cost CNY0.1~1 for 100 holes). In this case an alternative perforation technology would use the laser instead, but the cost is extremely higher (Approximately cost CNY10~15 for 100 holes). This implies new and cheaper perforation technologies for smaller holes are worth to be invented, or alternatively, new absorption structures with bigger perforated holes and better absorption in comparison with that of the MPP are deserved to be developed. This manuscript is an effort for this purpose. See reference by Bilong Liu and Xin Li:“ NOISE TRANSMISSION AND ABSORPTION OF LIGHTWEIGHT STRUCTURES: AN OVERVIEW AND EXPERIENCE”, Distinguished Plenary Lectures, ICSV26, Montreal, 7-11 July 2019 

Query: The terminology used eg. MPP, PPETs etc should be given systematically and should be clarified giving design parameters.How can be low frequencies measured in an impedance tube?

Reply:  Yes. The terminology used eg. MPP, PPETs etc has been revised and clarified. The low frequencies measured in an impedance tube is according to ISO 10534-2, which has been added and explained in section 4.

Query: How are these data reported to the scale of a panel that is used in-situ or commercialized?

 Reply: Theoretically, the performance of this sort of absorption structure doesn’t depend on the size of sample. However, the data reported here is in the impedance tube and therefore is only valid for normal incident waves. In-situ, the incident waves is more complicated and may come from many directions. This causes discrepancies in comparison with that of the data measured in the impedance tube. Still the data measured in the impedance tube has a good reference for the in-situ performance.

Query: There are a number of studies that report significant differences at low frequencies.

Reply: Yes. The differences at low frequencies in the report may be caused by improper experimental operation or machining size error. After repeated testing and correction, the measurement result is in good consistency with the theoretical result. The report has been revised, as shown in Figure6 and Figure7.

Query: A clear discussion and conclusions of the results is missing. 

Reply: Section3.2, Section4 and Section5 has been revised and conclusions of the results has been clearly discussed.

Query: References should be written with more accuracy.

Reply: The references have been modified in accordance with the prescribed format.

Reviewer 3 Report

Good paper, well written and clear in set-up and conclusions. 

Paper reads well, from the introduction through the setup of the experiments and the conclusions. 

Minor editing in English and a final proof-read suggested. 

One point for care and to make paper even more understandable: please make sure that all abbreviations are explained, either in a list or first time the abbreviation is used. This should include MPP, PPET (done, but perhaps location not ideal, as this should be closer to PPA), PPA. 

One additional suggestion - but perhaps for future research and papers: the focus of this study is low-frequency absorption and this is the most complicated frequency range. But it is a bit of a pity that all graphs stop at 500Hz or lower and that the frequency range above is never even talked about. Would it be easy to create combined panel that exhibits moderate to good sound absorption in all frequencies above 500Hz as well???

Author Response

General reply to three reviewers´ comments:

In our opinion the merits of our manuscript are:

1) The designed absorber introduces multiple resonances to expand the absorption bandwidth of more than one octave. In the restricted cavity space, the bandwidth of the proposed structure is obviously wider than that of MPP. Besides, the structure proposed avoids the extend tubes in comparison with that of PPETs when the similar sound absorption performance is concerned.

2)  The aperture of the structure designed for low frequency sound absorption is much larger in comparison with the MPP and therefore has much lower perforation numbers.

3)  The depth of each sub-cavity of the designed structure has the same size, which is convenient for production in practice.

Changes made:

1) To illustrate the potential of the extended bandwidth of the design structure, Section 3.2 has been revised and expanded.

2) In order to show that the design structure can be applied to different frequencies, Section 3.3 has been added.

3) A section discussing “how the panels could be made in practice, which materials could be used” been added to Section 4, as lines: 205-210.

4) Low-frequency absorption measurement in an impedance tube has been added in Section 4, as lines: 211-217, and the setup diagram has been revised, as Figure5.

5) After repeated testing and correction, the measurement results are in good consistency with the theoretical results. Figure 6 and Figure7 has been revised.

6) The references have been revised.

Detailed response to comments by reviewer:

Query: One point for care and to make paper even more understandable: please make sure that all abbreviations are explained, either in a list or first time the abbreviation is used. This should include MPP, PPET (done, but perhaps location not ideal, as this should be closer to PPA), PPA. 

Reply: Thank you for your detailed advices. The abbreviations used have been explained. ‘MPP’ is the abbreviation of ’micro perforated pane’;   ‘PPAs’ is the abbreviation of ’perforated panel absorbers’; ’PPET’ is the abbreviation of’ perforated panel with extended tubes’.

Query: One additional suggestion - but perhaps for future research and papers: the focus of this study is low-frequency absorption and this is the most complicated frequency range. But it is a bit of a pity that all graphs stop at 500Hz or lower and that the frequency range above is never even talked about. Would it be easy to create combined panel that exhibits moderate to good sound absorption in all frequencies above 500Hz as well???

Reply: This advice is very much appreciated. Section 3.3Comparison of PPAs with different cavity depth”  has been added to illustrate that this designed structure(apertures≥1.5mm) can be applied to above 500Hz by adjusting the depth of the back cavity.

Round 2

Reviewer 2 Report

I appriciate the efforts made by the authors in further improving the work presented in this paper.